# SMART PLACEMENT ENHANCED VISION: ENHANCING 3D-DETECTION WITH LEARNED 3D PLACEMENT

## ABSTRACT

The diversity and scale of annotated real-world 3D datasets limit the performance of monocular 3D detectors. Although data augmentation holds potential, creating realistic, scene-aware augmentations for outdoor environments presents a significant challenge. Existing augmentation methods majorly focus on realistic object appearance by advancing the rendering quality. However, we show that object placement is equally important for downstream 3D detection performance. The main challenge, however, for realistic placement, is to automatically identify the plausible physical properties (location, scale, and orientation) for placing objects in real-world scenes. To this end, we propose Smart-Placement, a novel 3D scene-aware augmentation method for generating diverse and realistic augmentations. In particular, given a background scene, we train a placement network to learn a distribution over plausible 3D bounding boxes. Subsequently, we render realistic cars from 3D assets and place them according to the locations sampled from the learned distribution. Through extensive empirical evaluation on standard benchmark datasets - KITTI and NuScenes, we show that our proposed augmentation method significantly boosts the performance of several existing monocular 3D detectors, setting a new state-of-the-art benchmark, while being highly data efficient.

## 1 INTRODUCTION

Monocular 3D object detection has rapidly progressed recently, enabling its use in autonomous navigation and robotics Huang et al. (2022); Ma et al. (2021). However, the performance of 3D detectors relies heavily on the quantity and quality of the training dataset. Given the considerable effort and time required to curate extensive, real-world 3D-annotated datasets, specialized data augmentation for 3D object detection has emerged as a promising direction.

However designing realistic augmentations for 3D tasks, is non-trivial, as the generated augmentations must adhere to the physical constraints of the real world, such as maintaining 3D geometric consistency and handling collisions. Existing techniques Ge et al. (2024); Lian et al. (2022) for 3D augmentation use relatively simple heuristics for placing synthetic objects in an input scene. For instance, in the context of road scenes, a recent approach Li et al. (2023) generates realistic cars and places them on the segmented road region. However, such heuristics result in highly unnatural scene augmentations (Fig. 1), resulting in a marginal improvement in 3D detection performance. In this work, we ask the following two crucial questions: (1) *What key factors are essential for generating realistic augmentations to improve monocular 3D object detection?*, and (2) *How can these factors be integrated to generate effective scene-aware augmentations?*

For the first question, we discover two *critical factors* responsible for generating effective 3D augmentations:

**1. Object Placement:** Plausible placement of augmented objects, with appropriate *physical properties (location, scale, and orientation)*, is essential for rendering realistic scene augmentations. For instance, in road scenes, a car should be placed on the road, be of appropriate size based on the distance from the camera, and follow the lane orientation. Augmentations that respect such physical constraints generalize better to real scenes by faithfully modelling the true distribution of the vehicles in the real world. To give an example of how such an augmentation looks, we compare our proposed augmentation approach against heuristic-based placement from Lift3D Li et al. (2023) in Fig. 1. Given the same rendering, our generation looks much more plausible regarding car place-

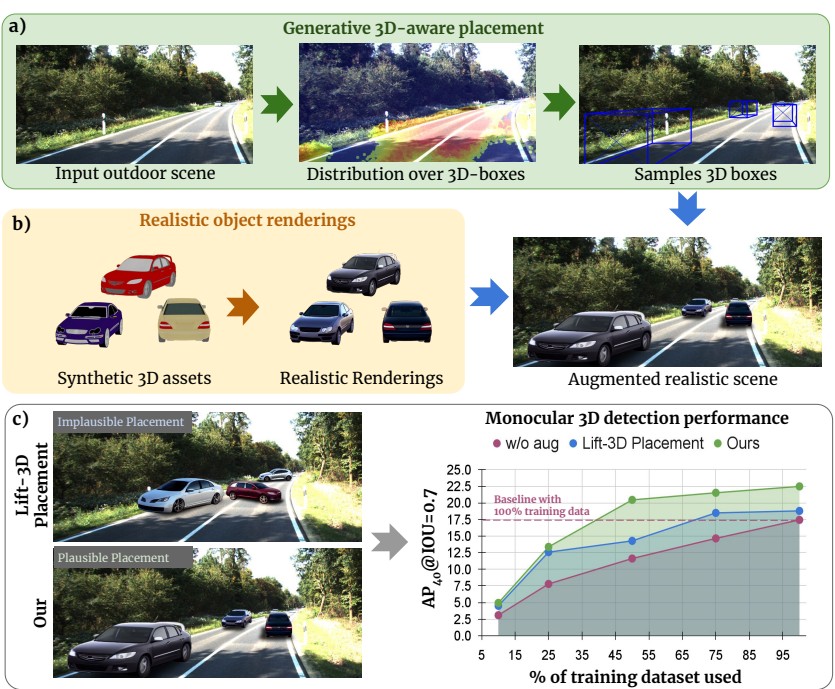

Figure 1: a) We compare augmentations from our learned placement with heuristic-based placements from Lift3D Li et al. (2023). In our augmentations, vehicles follow the lane orientations and are placed appropriately. b) These realistic augmentations significantly improve the 3D detection performance (KITTI Chen et al. (2015) val set, (easy)). Notably, we achieve detection performance comparable to that of the fully labeled dataset using only 50% of the dataset. Please refer to Appendix.A.5 for a detailed analysis.

ment and orientation compared to the baseline approach. Notably, when used for object detection training, our approach leads to significantly greater performance improvement, making the detector not only *performant*, but also *highly data efficient* (refer Fig. 1c)

**2. Object Appearance:** For 3D augmentation, it is desired that the generated objects exhibit realism and seamlessly integrate with the background to preserve visual consistency. This, in turn, minimizes the domain disparity between real and augmented data. Existing augmentation methods for 3D detection Li et al. (2023); Ge et al. (2024); Lian et al. (2022) primarily focus on the object appearance. This limits their ability to exploit the full potential of the data augmentations for 3D detection.

To address both these factors, we propose *Smart Placement*, a novel *scene-aware* augmentation method that generates effective 3D augmentations, as shown in Fig. 1. For plausible object placement, we train a 3D Scene-Aware Placement Network (**SA-PlaceNet**), which maps a given scene image to a distribution of plausible 3D bounding boxes. It learns realistic object placements that adhere to the physical rules of road scenes, facilitating sampling of diverse and plausible 3D bounding boxes (see Fig. 1a). For training this network, we consider existing 3D detection datasets, which typically contain only a limited number of objects per scene, resulting in a *sparse* training signal. Therefore, to enable *dense* placement prediction, we introduce novel modules based on (1) geometric augmentations of 3D boxes, along with (2) modeling of a continuous distribution of 3D boxes.

For realistic object appearance, we propose a rendering pipeline that leverages synthetic 3D assets and an image-to-image translation model. We translate the synthetic renderings into a realistic version using ControlNet Zhang & Agrawala (2023)(see Fig. 1b) and blend them with the background to get final augmentations. This allows us to utilize amateur-quality 3D assets and transform them into diverse, highly realistic car renderings that resemble real-world scenes.

Our two-stage augmentation approach is *highly effective and modular*, allowing seamless integration with advancements in placement and rendering for enhancing 3D object detection datasets. Using

our augmentation method on popular 3D detection datasets led to significant improvements over the prior baselines and set a new state-of-the-art monocular detection benchmark. Notably, as shown in Figure 1, using only 40% of the real training data and our 3D augmentations outperforms a model that is trained on the complete data without any 3D augmentations. Through extensive ablation studies, we thoroughly analyze the role of different components and their effect on detection performance. We summarize our contributions below:

1. We identify the critical role of *3D-aware object placement* and *realistic appearance* for generating effective scene augmentations for 3D object detection.

2. We propose *Smart-Placement*, a novel approach to generate plausible 3D augmentations for road scenes by realistically placing objects following scene grammar.

3. We demonstrate the effectiveness of the proposed augmentations on multiple 3D detection datasets and detector architectures with significant gains in performance as well as data efficiency.

## 2 RELATED WORK

**Object Placement.** There are numerous works Zhang et al. (2020); Zhu et al. (2023); Arroyo et al. (2021); Paschalidou et al. (2021); Wei et al. (2023) which aim to predict object placement by learning a transformation or the bounding box parameters directly for a given background image. Paschalidou et al. (2021); Wei et al. (2023) learns the distribution of indoor synthetic objects. Sun et al. (2020); Lee et al. (2018) learns the plausible locations for humans and other outdoor objects in a 2D manner. Few works aim to learn the arrangement conditioned on the scene-graph Luo et al. (2020); Jyothi et al. (2019); Yang et al. (2021). Zhang et al. (2020); Sun et al. (2020); Lee et al. (2018) are train a deep network adversarially in order to learn plausible 2D bounding box locations. Similarly, ST-GANLin et al. (2018) learns to predict the geometric transformation of a bounding box in the given scene using adversarial training. Li et al. (2019) uses a variational autoencoder to predict a plausible location heatmap over the scene but is limited to placement in restricted indoor environments.

**Monocular Object Detection** The current monocular 3D detection methods can be grouped as image-based or pseudo-lidar-based. Image-based detectors Brazil & Liu (2019); Liu et al. (2020); Mousavian et al. (2016); Roddick et al. (2018); Simonelli et al. (2019c;a); Wang et al. (2021); Liu et al. (2021); Zhang et al. (2021) estimate the 3D bounding box information for an object from a single RGB image. Due to the lack of depth information, these methods rely on geometric consistency in order to predict the class and the location of the object. Some works Li et al. (2020); Liu et al. (2020); Ma et al. (2021) use the prediction of key points of 3D bounding boxes as an intermediate task in order to improve it's performance on 3D monocular detection. In this work, we aim to improve the performance of image-based monocular detection models since RGB images are the most commonly used modality and easy to acquire with low acquisition costs, unlike LIDAR and depth sensors.

**Scene Data Augmentation.** Multiple works use 2D data augmentation techniques to improve the performance of perception tasks Shorten & Khoshgoftaar (2019). However, these augmentations cannot be lifted directly to 3D without violating the geometric constraints. To alleviate this problem, a recent method augments the training dataset for the task of 3D monocular detection Li et al. (2023); Lian et al. (2022); Tong et al. (2023a); Dokania et al. (2022). Lian et al. (2022) learns to paste cars on roads using the copy-paste operation by considering the cars' relative scale and pose. An interesting approach is taken by Dokania et al. (2022), where they model a synthetic urban scene from real-world distributions using available annotations to mimic the semantic properties of the real world. Li et al. (2023) learns a neural radiance field to generate realistic 3D cars with GAN augmented views. Tong et al. (2023a) learns the location to place 3D cars, but the placed cars look unrealistic. All these methods use heuristics such as lane segments to place cars; however, we aim to learn the distribution over car locations, scale, and orientation from the real-world object detection dataset.

## 3 METHOD
In this section, we first explain why it's important to have specialized methods for creating realistic scene-based augmentations for 3D detection. Then, we delve into the details of our unique approach to 3D augmentation.

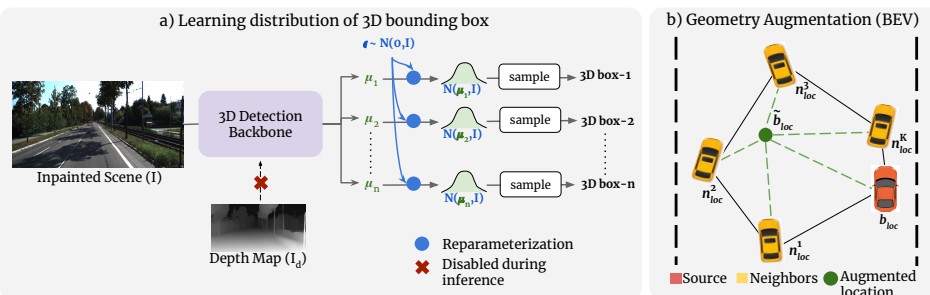

Figure 2: **a) SA-PlaceNet Architecture:** Given an input background image and corresponding depth to predict the means of a multi-dimensional Gaussian distribution over 3D bounding boxes. 3D bounding boxes are sampled from each of these Gaussian to compute the training loss. **b) Geometry-aware augmentation** in BEV (Birds Eye View). For a given source car location ($b_{loc}$), we first find $K$ nearest neighbors with the same orientation and augment the location to $\tilde{b}_{loc}$ by interpolating with neighboring locations $n_{loc}$ (Alg.3.1)

*Insight-1*: *Unlike the object-based augmentations suitable for broad image classification tasks, enhancing structured tasks as 3D object detection requires careful consideration of object-background and object-object interactions for generation of plausible scene-based augmentations.*

**Remarks:** Synthetic *object-based* augmentation for image classification typically involves placing objects on any suitable background. This method may not always respect the interaction between the object and the background, its impact on the classification task remains minimal. In contrast, for *scene-based* augmentation, which is crucial in tasks like 3D detection, the interactions between objects and backgrounds, as well as between objects, becomes pivotal. For example, implausible placements such as a car in a sky background, two cars occluding each other's 3D volume, or a car-oriented perpendicular to lanes on the road, need to be avoided. While one might argue that random placement could aid in a 3D object detection task by helping the model distinguish objects from the background, empirical evidence suggests otherwise. Hence, it's crucial to devise a placement-based augmentation method that respects the scene-prior, thereby instilling this understanding into the detector model during training.

*Insight-2*: *The distribution of augmented samples for a given real sample $\mathbf{x_r}$, denoted as $q(\mathbf{x_{aug}}|\mathbf{x_r})$, can be enhanced by better scene-prior modeling; this leads to augmented scenes that closely align with the real distribution, fostering a robust model that is resilient to failures and can achieve superior performance with fewer real samples.*

**Remarks:** The equation $q(\mathbf{x_{aug}}|\mathbf{x_r}) = q(\mathbf{x_{aug}}|\mathbf{z}, \mathbf{x_r})q(\mathbf{z}|\mathbf{x_r})$ represents the distribution of augmented samples for a given real sample $\mathbf{x_r}$. Here, $q(\mathbf{x}|\mathbf{z}, \mathbf{x_r})$ represents a pipeline that generates the augmented scene image upon applying an effective placement-based augmentation. Here, $q(\mathbf{z}|\mathbf{x_r})$ denotes the *scene-prior* related latent factor $\mathbf{z}$ given the real image. This factor can model the distribution of plausible location, orientation, and scale to place objects given the scene layout. Improved modeling of the *scene prior* ensures that the augmented scene closely matches the real distribution. Training with such augmentations imbues the model with a strong understanding of the *scene prior*, enhancing its robustness and reliability. We demonstrate that this strategy enables efficient training, yielding superior performance with fewer real samples compared to the baseline.

**Approach overview.** Our method for 3D augmentation consists of two stages. First, we train the placement model that maps a monocular RGB image to a distribution over plausible 3D bounding boxes (Sec. 3.1). Subsequently, we sample a set of 3D bounding boxes from this distribution to place cars. In the second stage, we render realistic cars following the sampled 3D bounding box and blend them with the background road scene. (Sec. 3.2).

## 3.1 SCENE-AWARE PLAUSIBLE 3D PLACEMENT

Realistic 3D placement in road scenes is extremely challenging due to the high diversity in the scene layouts and underlying *grammatical* rules of the road scenes (Sec.1). Existing methods use simple heuristic placement Li et al. (2023) based on the road segmentation unable to model these complexities and hence result in unnatural augmentations (Fig. 1). We propose a data-driven approach to

---

**Algorithm 1**: Procedure for geometric aware augmentation

---

1. **Input**:
   query box: $\mathbf{b} = [b_x, b_y, b_z, b_h, b_w, b_l, b_\theta, b_\alpha]$ where $b_{loc} = (b_x, b_y, b_z)$
   number of neighbors: $\mathbf{K}$
   radius of interpolation: $\mathbf{r}$
   amount of jitter: $\mathbf{d_j}$
   orientation threshold: $\epsilon_\theta$

2. Sample $K$ neighbors $\{n^i\}_1^K \in B$, **s.t.**

$$||n_{loc}^i - b_{loc}||_2 < r \quad \& \quad |n_\theta^i - b_\theta| < \epsilon_\theta \tag{1}$$

3. **If** there are no neighbours i.e $K = 0$, **then do**

$$b_x \leftarrow b_x + d_x \quad b_z \leftarrow b_z + d_z \tag{2}$$

   where $d_z > 2d_x$ and $d_x, d_z \in \mathcal{U}(0, d_j)$
   **end If**

4. **Else do**
   Generate the augmented location $\tilde{b}_{loc} = (\tilde{b}_x, \tilde{b}_y, \tilde{b}_z)$ using Eq. 7
   **end Else**

5. **Output** : Augmented bounding box parameters $\tilde{b} : [\tilde{b}_x, \tilde{b}_y, \tilde{b}_z, b_h, b_w, b_l, b_\theta, b_\alpha]$

---

learn the real-world placement distribution by training a **Scene-Aware Placement Network (SA-PlaceNet)**, that maps a given image to the distribution of plausible 3D bounding boxes.

Learning such a distribution requires dense supervision about object location, scale, and orientation for each 3D point in space. Having such a dense annotated real dataset is impractical and can only be generated in a controlled synthetic setting that does not generalize to the real world. Hence, we take an alternate approach to learn the 3D bounding box distribution from an existing 3D object detection dataset. Object detection datasets only provide information on where *cars are located* but not *where they could be*. To mitigate this, we inpaint the vehicles from the scene to generate a paired image dataset with/without the vehicles. However, detection datasets have only a few vehicles in each scene, which provides only *sparse* signals for plausible 3D bounding boxes. Directly training with such a dataset will lead to overfitting and the model learns the *sparse* point estimate of locations as each scene has only a few car locations in the ground truth. To truly learn the underlying distribution of 3D bounding boxes, we propose two novel modules during training of placement network. **Geometry aware augmentation** and predicting *distribution over 3D bounding box* instead of a single estimate. The proposed modules enable diverse placements for a given scene that follow the underlying rules of the road scene.

The complete architecture for placement is shown in Fig. 2a. We build SA-PlaceNet using the backbone of MonoDTR Huang et al. (2022). MonoDTR is designed to perform monocular 3D object detection and is trained with auxiliary depth supervision. However, depth is not required during inference. We adapt the architecture of MonoDTR to learn the mapping from background road images $\mathcal{I}$ to a set of 3D bounding boxes $\mathcal{B}$. Following Huang et al. (2022), we define bounding box $\mathbf{b} \in \mathcal{B}$ as 8 dimensional vector $\mathbf{b} = [b_x, b_y, b_z, b_h, b_w, b_l, b_\theta, b_\alpha]$, where $(b_x, b_y, b_z)$ are 3D locations, $(b_h, b_w, b_l)$ are height, width, and length of the box, and $b_\theta$ and $b_\alpha$ are orientation angles. Note that $b_\alpha$ can be computed deterministically from $b_\theta$ and hence we have only 7 variables defining a given bounding box. As a convention, we consider the $xz$ plane as the road plane.

**Dataset preparation.** There is no existing real-world dataset that provides plausible placement annotations for a given road scene. Instead, we take advantage of the KITTI Geiger et al. (2013) dataset with 3D object detection annotations. We preprocess the dataset by inpainting the foreground cars in the scene using off-the-shelf inpainting Rombach et al. (2022). Through this process, we obtain an image dataset ($\mathcal{I}$) with no cars on the road and a set of corresponding 3D bounding boxes ($\mathcal{B}$). Next, we obtain depth images $\mathcal{I}_d$ for the inpainted images using Ranftl et al. (2021). The obtained paired dataset, $\mathcal{D} = \{\mathcal{I}, \mathcal{I}_d, \mathcal{B}\}$, is used to train the SA-PlaceNet.

### 3.1.1 GEOMETRY AWARE AUGMENTATION

Training SA-PlaceNet directly with the paired dataset $\mathcal{D}$ could easily learn a mapping to sparse 3D locations where real cars were present before inpainting. Additionally, the model can cheat by using

Figure 3: **Rendering pipeline:** Given a 3D asset, we first render an image and shadow from a fixed light source according to the 3D box parameters. Next, we used edge-conditioned ControlNet Zhang & Agrawala (2023) to generate a realistic car version that follows the same orientation and scale as the rendered image. Finally, we use the obtained shadow, rendered car, and 3D location to place the car and render augmented images.

the inpainting artifacts to predict cars at the source location. To overcome these limitations, we propose *geometry-aware augmentation* $\mathcal{G}$ in the 3D bounding box space. We build on the intuition that the regions' neighboring ground truth car locations are also plausible for placement. The augmentation $\mathcal{G}$ transforms the ground truth bounding box $\mathbf{b} \in \mathcal{B}$ of a car, located at $\mathbf{b_{loc}} = (b_x, b_y, b_z)$ into a plausible neighboring box $\tilde{\mathbf{b}} = \mathcal{G}(\mathbf{b})$ located at $\tilde{\mathbf{b}}_{\mathbf{loc}} = (\tilde{b}_x, \tilde{b}_y, \tilde{b}_z)$ shown in Fig. 2b. The detailed algorithm for geometry-aware augmentation is given in detail in Alg.3.1. Specifically, we first find a set of $K$ neighboring car boxes $\{n^i\}_{i=1}^{i=K}$ to the given car $\mathbf{b}$. We consider $n^i$ as the neighbor of $\mathbf{b}$ if $||n_{loc}^i - \mathbf{b_{loc}}||_2 < r$ and $|n_\theta^i - b_\theta| < \epsilon_\theta$, for a given threshold $r$ and $\epsilon_\theta$. We assume the selected $K$ nearest cars will be in the same lane and follow similar orientations. To augment the location $\mathbf{b_{loc}}$, we take a convex combination of neighboring locations $n_{loc}^i$ and $\mathbf{b_{loc}}$ and obtain a location $\tilde{\mathbf{b}}_{\mathbf{loc}}$.

$$\tilde{b}_{loc} = \lambda_0 * b_{loc} + \sum_{i=1}^{k} \lambda_i * n_{loc}^i \tag{3}$$

where $\sum_i \lambda_i = 1$, $\lambda_i \geq 0$ are hyperparameters randomly sampled for each ground truth box $\mathbf{b}$. This transformation enables us to span a large region of plausible locations during training, hence enabling diverse placement locations during inference for each scene. If a car doesn't have any neighboring cars, we apply a uniform jitter along the length and a smaller jitter along the width of the car bounding box.

### 3.1.2 DISTRIBUTION OVER 3D BOUNDING BOXES.

Geometry-aware augmentation enables the generation of diverse placement locations, but it learns a direct mapping from the input image to a point estimate of bounding boxes. To learn a continuous representation in the output space, we map the input image to the distribution of 3D boxes. This improves the coverage of plausible locations and enables diverse bounding box sampling from a predicted set of mean boxes. Specifically, we approximate each predicted bounding box $\mathbf{b}$ as a multi-dimensional Gaussian distribution with mean $\mu_b$ and a fixed covariance matrix as $\alpha I$, where $\alpha$ is used to control the spread as shown in Fig. 2a. We empirically observed that having a fixed covariance improves training stability. Having a higher $\alpha$ value results in strong augmentations, where the sampled car is far away from the mean location, resulting in a weaker training signal. During the forward pass, the SA-PlaceNet predicts mean bounding box parameters $\mu_b$. To sample a box $\hat{\mathbf{b}}$, we first sample $\epsilon \in \mathcal{N}(\mathbf{0}, \mathbf{I})$ and use the reparametrization trick as follows:

$$\hat{\mathbf{b}} = \mu_b + \epsilon * \alpha \mathbf{I} \tag{4}$$

### 3.1.3 SA-PLACENET TRAINING.

We train SA-PlaceNet with the acquired paired dataset $\mathcal{D} = \{\mathcal{I}, \mathcal{I}_d, \mathcal{B}\}$, consisting of inpainted background image ($\mathcal{I}$), inpainted depth image ($\mathcal{I}_d$) and the ground truth 3D bounding boxes ($\mathcal{B}$). Following Huang et al. (2022), we train the model with $\mathcal{L}_{cls}$ for objectness and class scores, $\mathcal{L}_{dep}$ for depth supervision, and $\mathcal{L}_{reg}$ for bounding box regression.The proposed modules for *geometry-aware augmentation* and *learning distribution over 3D bounding boxes* can be easily integrated into a modified version of the regression loss $\mathcal{L}_{reg}^m$ as discussed below. The total loss is then defined as:

$$\mathcal{L} = \mathcal{L}_{cls} + \mathcal{L}_{reg}^m + \mathcal{L}_{dep} \tag{5}$$

For a given ground-truth bounding box parameter $\mathbf{b}$, we first augment it using geometry-aware augmentation following Eq. equation 7 to obtain modified bounding box parameters $\tilde{\mathbf{b}} = \mathcal{G}(\mathbf{b})$. To capture the distribution of 3D boxes, we predict a mean bounding box parameter $\mu_b$ instead of a point estimate of the box parameters and randomly sample a new bounding box $\hat{\mathbf{b}}$ using the reparameterization trick outlined in Eq. equation 4. Subsequently, we compute the modified regression loss between the model prediction $\mu_b$ and the ground truth box $\mathbf{b}$ as follows:

$$\mathcal{L}_{reg}^m(\mu_b, \mathbf{b}) = \mathcal{L}_{reg}(\hat{\mathbf{b}}, \tilde{\mathbf{b}}) \qquad (6)$$

### 3.2 WHAT TO PLACE? RENDERING CARS

We generate realistic scenes by selecting cars and rendering them within the projected 3D coordinates of the predicted location, as shown in Fig. 3. To accurately render a car based on 3D bounding box parameters, we utilize 3D car assets from ShapeNet Chang et al. (2015) that can be adjusted through orientation and scale transformations. Upon acquiring the 3D bounding box predictions, our rendering step entails sampling cars from the ShapeNet. Subsequently, the car model undergoes rotation according to the 3D observation angle of the object before positioning it within the designated scene. We separately render car shadows with predefined lighting in the rendering environment, following Chen et al. (2021). The rendered ShapeNet car images, although following the 3D bounding boxes, look unrealistic when pasted into the scene (Fig. 6, row-2). To resolve this, we leverage the advances in conditional generation using text-to-image models.

For the generated synthetic car images, we apply an edge detector to obtain an edge map. The edge map preserves the car's structure and still follows the same orientation and scale as the original car. Next, we use edge-conditioned text-to-image diffusion model ControlNet Zhang & Agrawala (2023) to render a realistic car using the prompt '*A realistic car on the street.*' We further finetune the

backbone diffusion model in ControlNet using LoRA Hu et al. (2022) on a subset of 'car' images from the KITTI dataset. This enables us to generate natural-looking versions of cars that blend well with the background scene (Fig. 6). As ControlNet enables diverse generations from the same edge image, we can generate multiple renderings of cars from the edge map of a single ShapeNet car. This enables the generation of many diverse cars from a small, fixed set of 3D assets. The generated renderings look realistic and substantially boost object detection performance, as shown in Tab. We believe, the proposed approach of using a few 3D assets with conditional text-to-image models is promising and can be applied to generate diverse 3D augmentations for other tasks as well. Apart from the proposed rendering technique, we also experiment directly placing ShapeNet Chang et al. (2015) and renderings

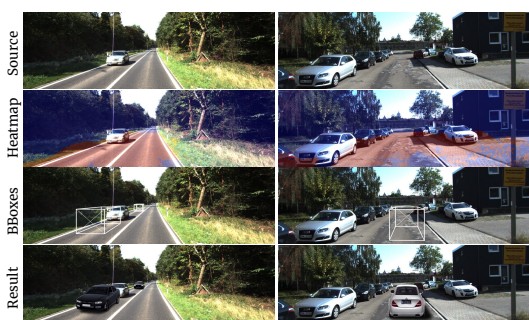

Figure 4: Given an input source image, we plot the heatmaps of the mean objectness score at each pixel location. The generated heatmaps span a large region on the road with plausible locations of objects. Next, we show samples of bounding boxes and realistic renderings of cars in the scene.

from Lift3D Li et al. (2023), which is a generative radiance field approach that generates realistic 3D car assets.

## 4 EXPERIMENTS

In this section, we present results for 3D-aware placement (Sec. 4.1) and car renderings (Sec. 4.2). Next, we present the results of 3D object detection when trained with our generated augmentations (Sec. 4.3). We show additional results on 2D detection, additional ablations, and quantitative analysis of SA-PlaceNet in the suppl. document.

**Dataset.** We use the KITTI Geiger et al. (2013) and NuScenes Caesar et al. (2019) datasets for our experiments. KITTI consists of a total of 7481 real-world images captured from a camera mounted on a car. Following Li et al. (2023); Tong et al. (2023b); Chen et al. (2015), we split the data into 3712 train and 3679 validation images. For NuScenes, we use the official split with 700 train scenes containing 28130 images and 150 validation scenes containing 6019 images.

## 4.1 EVALUATION OF PLACEMENT MODEL

The placement network is trained with RGB images from the train split. We prepare the training data by inpainting the moving objects using Rombach et al. (2022) and obtain a paired dataset $\mathcal{D} = \{\mathcal{I}, \mathcal{I}_d, \mathcal{B}\}$ as detailed in Sec. 3.1. To visualize the performance of the placement, we generate heatmaps over the center of the bottom face of the bounding box in Fig. 4. For visualization, we use the mean objectness score of the anchor boxes corresponding to each grid cell. Geometry-aware augmentation enables learning of a large region for placing cars even though trained with input scenes with only a few cars. This allows for the sampling of diverse physically plausible placement locations for a given input scene shown as a set of 3D bounding boxes. We sample two sets of boxes from the predicted distribution. The sampled boxes have appropriate locations, scales, and orientations based on the background road. We present a detailed quantitaitve analysis of our method in Appendix A.1.1.

**Analysis.** We analyze the impact of each component on placement performance in Fig. 5a). The naive baseline of directly training object placement without geometric augmentation and variational modeling only learns a point estimate and results in a few concentrated spots for placement location. Adding the variational head for learning a distribution of boxes instead expands the space of plausible locations but is still segregated in small regions. For the variational

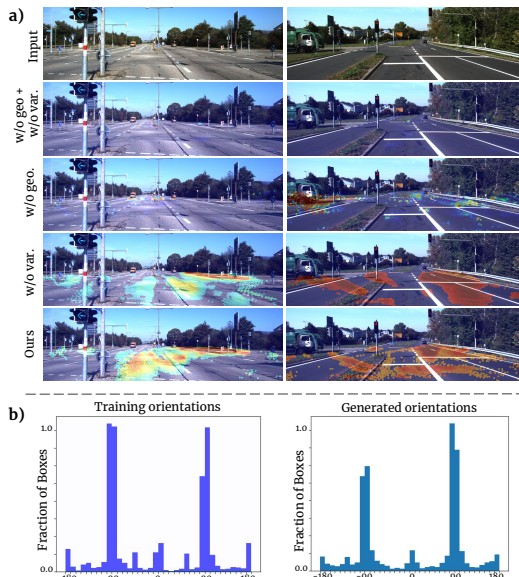

Figure 5: a) Qualitative comparison for object placement - For a background road scene image, we visualize the heatmaps of aggregated objectness scores at each pixel location. Our proposed method is capable of predicting dense regions on the road that are semantically plausible for placing cars. b) Histogram of the distribution of orientations of the ground truth bounding boxes and the generated bounding boxes.

head, we have fixed the $alpha$ as 0.1. This highlights the sparse training signals for placement using ground truth boxes. However, when coupled with the geometry-aware augmentation, the predicted distribution covers a large driveable area on the road. To further analyze the orientations, we plot a histogram of predicted and the ground truth orientations in Fig. 5b), where the predictions closely follow the ground truth.

## 4.2 RENDERING OBJECTS

We augment the road scenes by placing synthetic cars rendered by several approaches in Fig. 6. We compare the rendering quality of the proposed method with **1) ShapeNet** - 3D car assets renderings sampling from ShapeNet Chang et al. (2015), **2) Lift3D** Li et al. (2023) - A generalized NeRF method for generating 3D car models. ShapeNet renderings result in unnatural augmentations due to synthetic car appearance and domain gaps from real scenes. On the other hand, Lift3D renderings, although realistic, lack diversity and suffer from artifacts. Our rendering method leverages conditional text-to-image diffusion models and generates extremely re-

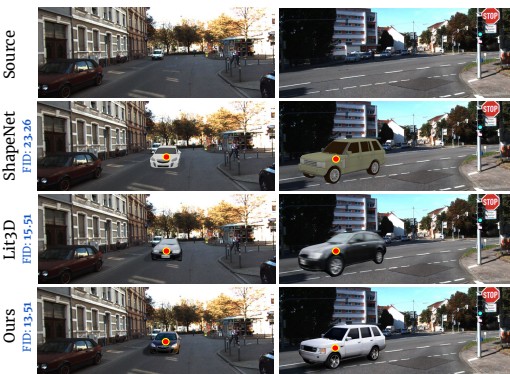

Figure 6: Ablation over rendering methods: Given the source image and predicted 3D bounding boxes, we sample and render a synthetic ShapeNet Chang et al. (2015) car; Lift3D Li et al. (2023) rendered method; and our realistic rendering. Observe that the cars in our rendering match the scene lighting conditions well. This is due to the smaller domain gap of the rendered cars with the training samples.

alistic cars that blend well with the background and are of high fidelity. Additionally, as our rendering starts from an underlying 3D asset, we use it to render shadows in a synthetic environment and copy the same shadow to the generated realistic renderings. The proposed rendering pipeline effectively generates realistic augmentations and results in superior object detection performance (Tab. 1). Further, we report FID of the generated augmentations with the real training set to evaluate the realism.

### 4.3 ENHANCING 3D OBJECT DETECTION PERFORMANCE

We evaluate the effectiveness of our augmentations for monocular 3D object detection. We augment the training set with the same number of images to prepare an augmented version of the dataset. We compare our proposed augmentation method with the following augmentation approaches:

**Geometric Copy-paste (Geo-CP) Lian et al. (2022).** We use instance-level augmentation from Lian et al. (2022), where cars from the training images are archived along with the corresponding 3D bounding boxes to create a dataset. For augmenting a scene, a car, and its 3D box parameters are sampled from the dataset and car is simply pasted in the background.

**Lift-3D Li et al. (2023)** proposed a generative radiance field network to synthetize realistic 3D cars. The generated cars are then placed on the road using a heuristic-based placement. Specifically, a placement location is sampled on the segmented road, and other 3D bounding box parameters are sampled from a predefined parameter distribution.

**CARLA Dosovitskiy et al. (2017).** To compare the augmentations generated by simulated road scene environments, we use state-of-the-art CARLA simulator engine for rendering realistic scenes with multiple cars. It can generate diverse traffic scenarios that are implemented programmatically. However, it's extremely challenging for simulators to capture the true diversity from real-world road scenes and they often suffer from a large sim2real gap.

**Rule Based Placement (RBP).** We create a strong rule-based baseline to show the effectiveness of our learning-based placement. Specifically, we first segment out the road region with Han et al. (2022) and sample placement locations in this region. To get a plausible orientation, we copy the orientation of the closest car in the scene, assuming neighboring cars

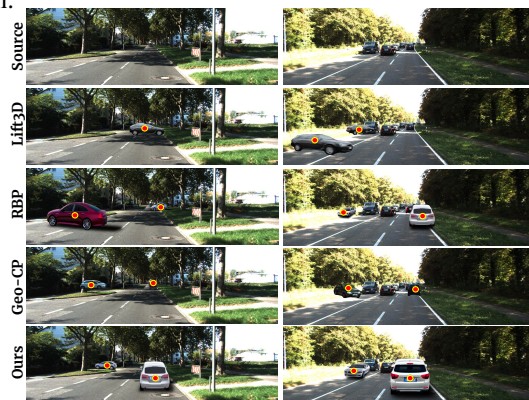

Figure 7: Qualitative comparison of the generated augmentations with all the baseline methods. Our augmentations are highly realistic, place cars following plausible physical properties, and have a minimal domain gap from the training dist.

follow the same orientations. We used our our rendering pipeline to generate realistic augmentations.

**Qualitative comparison of generated augmentations** are shown in Fig. 7. Lift3D augmentations have cars placed in incorrect orientation as the orientation is sampled from a general predefined distribution. RBP and Geo-CP augmentations are relatively better in terms of orientation but fail to place cars in the correct lanes. The proposed augmentation method follows the underlying grammar of the road well and generates realistic scene augmentations.

Table 1: Monocular 3D detection performance on KITTI dataset

| a) MonoDLE Ma et al. (2021) | 3D@IOU=0.7 | | | 3D@IOU=0.5 | | | b) GUPNet Lu et al. (2021) | 3D@IOU=0.7 | | | 3D@IOU=0.5 | | |
|---|---|---|---|---|---|---|---|---|---|---|---|---|---|
| | Easy | Mod. | Hard | Easy | Mod. | Hard | | Easy | Mod. | Hard | Easy | Mod. | Hard |
| w/o 3D Augmentation | 17.45 | 13.66 | 11.69 | 55.41 | 43.42 | 37.81 | w/o 3D Augmentation | 22.76 | 16.46 | 13.27 | 57.62 | 42.33 | 37.59 |
| Geo-CP | 17.52 | 14.60 | 12.57 | 58.95 | 44.23 | 38.66 | Geo-CP | 21.81 | 15.65 | 13.24 | 59.12 | 44.03 | 39.16 |
| CARLA | 17.98 | 14.30 | 12.17 | 58.33 | 44.41 | 38.81 | CARLA | 22.50 | 16.17 | 13.61 | 59.89 | 43.52 | 38.22 |
| Lift3D | 17.19 | 14.65 | 12.48 | 56.81 | 44.21 | 39.13 | Lift3D | 19.05 | 14.84 | 12.64 | 57.50 | 43.81 | 39.22 |
| RBP | 20.50 | 14.32 | 11.29 | 60.30 | 43.69 | 38.55 | RBP | 21.67 | 14.56 | 11.23 | 60.40 | 43.25 | 36.95 |
| Ours | **22.49** | **15.44** | **12.89** | **63.59** | **45.59** | **40.35** | Ours | **23.94** | **17.28** | **14.71** | **61.01** | **47.18** | **41.48** |

### 4.3.1 REALISTIC AUGMENTATIONS IMPROVES 3D DETECTION.

We evaluate our augmentation technique on two state-of-the-art monocular 3D detection networks - MonoDLE Ma et al. (2021) and GUPNet Lu et al. (2021) in Tab. 1 on KITTI Geiger et al. (2013) dataset. We generate one augmentation per real image for all the baselines. All the augmentation techniques improve over the baseline for MonoDLE. However, gains from Lift3D, CARLA, and Geo-CP are marginal. RBP performs better than other baselines primarily due to our realistic ren-

derings. For GUPNet, none of the baselines can improve the detection performance overall. Our proposed method significantly improves the score detection scores for both networks. This indicates a strong generalization of our augmentations on various 3D object detection models. We also show results on the current state-of-the-art MonoDETR Zhang et al. (2022) in Appendix A.4.1.

### 4.3.2 IMPACT OF RENDERING FOR 3D OBJECT DETECTION.

Table 2 presents an ablation study of various rendering approaches for augmentation in 3D detection. All renderings, when used with our learned placement, significantly outperform the baselines, demonstrating their compatibility with any rendering method. ShapeNet shows the lowest performance due to limited synthetic

Table 2: Rendering ablation with fixed placement

| Rendering | 3D@IOU=0.7 | | | 3D@IOU=0.5 | | |
|---|---|---|---|---|---|---|
| | Easy | Mod. | Hard | Easy | Mod. | Hard |
| w/o 3D Augmentation | 17.45 | 13.66 | 11.69 | 55.41 | 43.42 | 37.81 |
| ShapeNet | 20.91 | 14.17 | 12.28 | 59.54 | 43.48 | 37.64 |
| Lift3D | 21.35 | 14.25 | 11.65 | 60.38 | 42.65 | 37.53 |
| Ours (w/o shadow) | 21.45 | 14.21 | 11.73 | 61.23 | 43.27 | 38.28 |
| Ours | **22.49** | **15.44** | **12.89** | **63.59** | **45.59** | **40.35** |

car diversity and a substantial sim2real gap. Lift3D rendering performs better than ShapeNet but exhibits noticeable artifacts when cars are close to the camera (Fig. 6). Our rendering approach, which uses a generative text-to-image model, outperforms all baselines but also enhances and achieves state-of-the-art performance when combined with shadows.

### 4.3.3 AUGMENTING OTHER CLASSES

Though the car is the major category in the road 3D detection benchmarks, we also perform augmentation for two additional categories of cyclists and pedestrians, given they occur at 3.79 % and 11.39 % in the KITTI training set. For simplicity, we integrate our placement method with copy-paste rendering as described in Appendix A.6.1 (similar to Geo-CP Lian et al. (2022)). Note that we trained another placement model to predict the placement of all the classes together. We use the augmented dataset with renderings of cyclists and pedestrians to train MonoDLE Ma et al. (2021) object detector. The results are shown in Tab. 3; our augmentation significantly improves the detection performance of both categories over the baselines. We show qualitative results for other classes in the Appendix. A.1.4.

Table 3: Augmenting multiple categories for 3D detection

| Cyclist | 3D@IOU=0.50 | | | 3D@IOU=0.25 | | | Pedestrian | 3D@IOU=0.50 | | | 3D@IOU=0.25 | | |
|---|---|---|---|---|---|---|---|---|---|---|---|---|---|
| | Easy | Mod | Hard | Easy | Mod | Hard | | Easy | Mod | Hard | Easy | Mod | Hard |
| w/o 3D Augmentation | 4.92 | 2.03 | 1.85 | 18.41 | 10.82 | 9.52 | w/o 3D Augmentation | 4.60 | 3.81 | 2.99 | 22.98 | 18.38 | 15.12 |
| Ours | **6.75** | **3.41** | **3.37** | **21.59** | **11.23** | **9.90** | Ours | **4.98** | **3.89** | **3.34** | **26.28** | **20.81** | **16.16** |

### 4.4 EXPERIMENTS ON LARGE DATASETS

We validate the generalization of our method by training SA-PlaceNet on a large driving dataset - NuScenes (Caesar et al., 2019). Our approach produces plausible realistic augmentations for the given scene (see Appendix A.1.3) and we show improved performance on the NuScenes dataset with the FCOS3D (Caesar et al., 2019) monocular detection network in Tab. 4.

Table 4: Detection on NuScenes

| FCOS3D Caesar et al. (2019) | MAP | NDS |
|---|---|---|
| w/o 3D Augmentation | 0.3430 | 0.415 |
| Lift3D | 0.3211 | 0.371 |
| Ours | **0.3704** | **0.440** |

### 4.5 COST OF SMART PLACEMENT

Training of SA-PlaceNet takes a fraction of the time of the detection training. The relative training time reduces significantly on the large datasets such as NuScenes. We present the computational requirements of our augmentation in comparison to the training time in Table 5. We train GUPNet and MonoDLE for an additional 10 epochs and FCOS3D for an additional 5 epochs when training with our augmented data as compared to the training on the original dataset.

Table 5: Analysis of Training Time

| Model | Dataset | Training Time | #GPU's | GPU Model |
|---|---|---|---|---|
| SA-PlaceNet | KITTI | 12h | 1 | A5000 |
| SA-PlaceNet | NuScenes | 32h | 1 | A5000 |
| GUPNet | Original KITTI | 20h | 1 | A5000 |
| GUPNet | Augmented KITTI | 22h | 1 | A5000 |
| FCOS3D | Original NuScenes | 5d18h | 2 | A5000 |
| FCOS3D | Augmented NuScenes | 6d | 2 | A5000 |

## 5 CONCLUSION

This work proposes a novel scene-aware augmentation technique to improve outdoor monocular 3D detectors. The core of our method is an object placement network, that learns the distribution of physically plausible object placement for background road scenes from a single image. We utilize this information to generate realistic augmentations by placing cars on the road scenes with geometric consistency. Our results with scene-aware augmentation on monocular 3D object detectors suggest that realistic placement is the key to substantially improving the augmentation quality and data efficiency of the detector. The primary limitation of our approach is the dependency on the off-the-shelf inpainting method for data preparation for the training of the placement network. Also, our current framework does not consider more nuanced appearance factors in augmentations such as the lighting of the scene. In conclusion, we provide important insights for designing scene-based augmentations for 3D object detection.

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

# A APPENDIX

## A.1 ADDITIONAL PLACEMENT RESULTS

### A.1.1 QUANTITATIVE EVALUATION

To quantify the performance of placement, we compute the following three metrics on the training set of KITTI: **1) Overlap:** As road regions can cover most of the plausible locations for cars, we evaluate the predicted location by checking whether the *center of the base of the 3D bounding box* is on the road. Specifically, we compute the fraction of boxes that overlap with the road segmentation obtained using Han et al. (2022). **2) $\theta_{\mathbf{KL}}$:** We evaluate the KL-divergence between the distribution of orientation of the predicted 3D bounding box and the ground truth boxes. We present quantitative results in Tab. 6, where our method achieves superior overlap scores, suggesting the superiority of placement.

Table 6: Ablation over SA-PlaceNet components

| Method | Random | w/o var & geo | w/o geo | w/o var | Ours |
|---|---|---|---|---|---|
| Overlap ↑ | 0.20 | 0.15 | 0.17 | 0.35 | **0.36** |
| $\theta_{KL}$ ↓ | 1.37 | 0.66 | 1.18 | 0.32 | **0.30** |

### A.1.2 CONTROLLING TRAFFIC DENSITY IN SCENES

Our augmentation method enables us to control the traffic density of vehicles in the input scenes by controlling the number of bounding boxes to be sampled. We present results for generating low-density ($1 - 3$ cars added) and high-density ($3 - 5$ cars added) traffic scenes in Fig. 8.

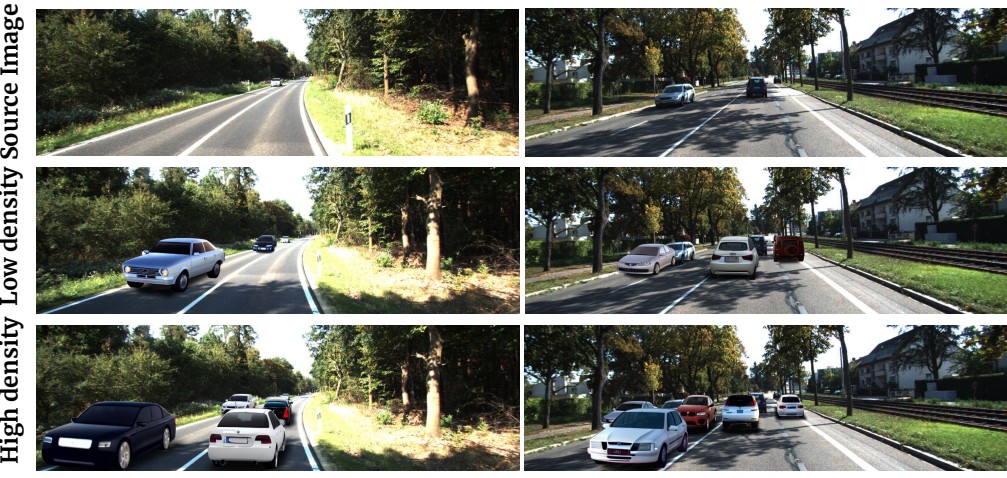

Figure 8: Augmented training dataset for 3D object detection: Given a sparse scene with few cars, we place cars at the predicted 3D bounding box locations using our rendering algorithm. We present two sets of results, one with low density ($1-3$ cars added) and another with high density ($4-5$ cars added) for each scene.

### A.1.3 PLACEMENT ON NUSCENES CAESAR ET AL. (2019) DATASET

We validate the generalization of our method by training SA-PlaceNet on a subset of a recent driving dataset - NuScenes Caesar et al. (2019) in Fig. 9. We visualize predicted 3D bounding boxes and realistic renderings from our method. Our approach produces plausible placements and authentic augmentations for the given scene.

Source Image  Sampled 3D boxes  Augmented Scene

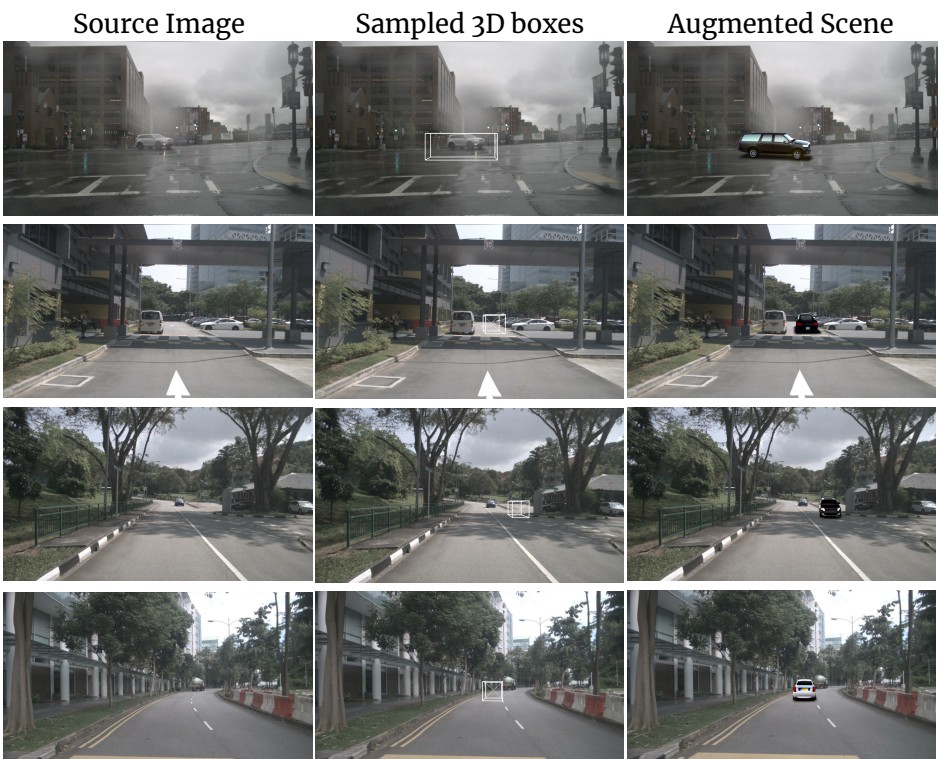

Figure 9: Placement on NuScenes Caesar et al. (2019) dataset.

### A.1.4 PLACING OTHER CATEGORIES

Our method enables us to learn placement for other categories from KITTI datasets. Specifically, we trained a joint placement model to learn the distribution of 3D bounding boxes for cars, pedestrians, and cyclists. To render the pedestrians and cyclists, we leverage simple copy-paste rendering as discussed in Sec. A.6.1. We present placement results in additional categories in Fig. 10. The proposed method predicts plausible locations, orientation, and shape of the object, enabling rich scene augmentations. Using these augmentations for training leads to significant improvement in performance for less frequent cyclist and pedestrian categories (Tab. 3 in the main paper).

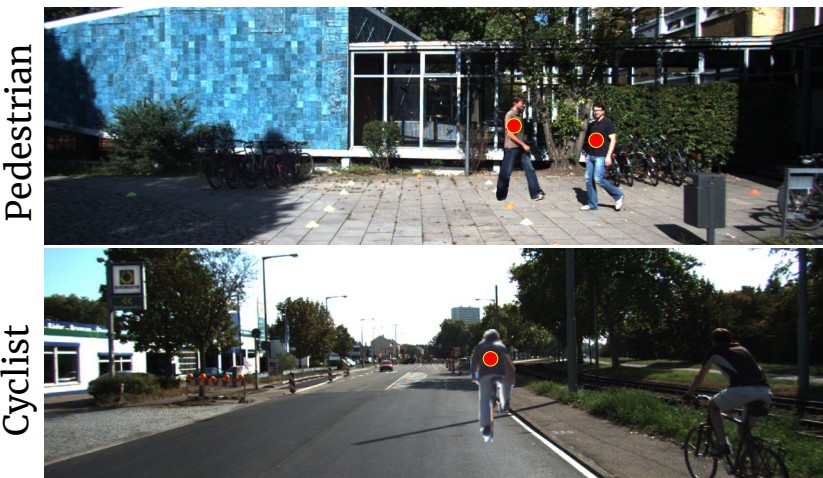

Figure 10: Placement results for other categories

## A.2 GENERALIZATION OF SA-PLACENET

To validate the generalization capability of our placement network, we infer our model trained on KITTI dataset on Vitual KITTI (VKITTI) dataset. Specifically, we use SA-Placenet to predict the placement locations in VKITTI and augment the images by placing new cars. We perform 3D monocular detection on VKITTI on a $50 - 50$ split for training and validation images to evaluate the generalization of our SA-PlaceNet in generating realistic augmentations for improving 3D detection. We show the results in Tab 7.

Table 7: Performance on VKITTI

| MonoDLE | 3D@IOU=0.7 | | | 3D@IOU=0.5 | | |
|---|---|---|---|---|---|---|
| | Easy | Mod. | Hard | Easy | Mod. | Hard |
| w/o 3D Augmentation | 15.78 | 11.67 | 8.71 | 48.98 | 38.18 | 30.11 |
| Ours | 17.71 | 12.21 | 8.90 | 50.19 | 39.78 | 30.91 |

## A.3 IMPLEMENTATIONS DETAILS

### A.3.1 PLACEMENT DATA PREPROCESSING

We use the state-of-the-art Image-to-Image Inpainting method Rombach et al. (2022) to remove vehicles and objects from the KITTI dataset Geiger et al. (2013). The input prompt *'inpaint'* is passed to the inpainting pipeline. A few outputs from this method can be seen in Fig. 11

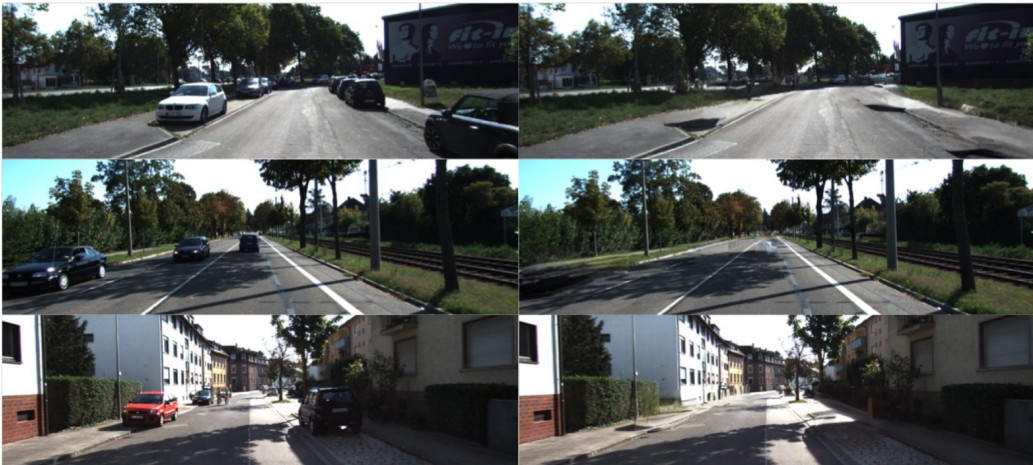

Source Image        Inpainted Image

Figure 11: Outputs generated from Stable Diffusion Inpainting pipeline Rombach et al. (2022). These inpainted images are used for training our placement model.

### A.3.2 BASELINE METHODS

**Geometric Copy-paste (Geo-CP).** To augment a given scene, a car is randomly sampled from the database, and its 3D parameters are altered before placement. Specifically, the depth of the box ($z$ coordinate) is randomly sampled, and corresponding $x$ and $y$ are transformed using geometric operations. Other parameters, such as bounding box size and orientation, are kept unchanged. The sampled car is then pasted using simple blending on the background scene.

**CARLA Dosovitskiy et al. (2017).** To compare the augmentations generated by simulated road scene environments, we use state-of-the-art CARLA simulator engine for rendering realistic scenes with multiple cars. It can generate diverse traffic scenarios that are implemented programmatically. However, it's extremely challenging for simulators to capture the true diversity from real-world road scenes and they often suffer from a large sim2real gap.

**Rule Based Placement (RBP).** We create a strong rule-based baseline to show the effectiveness of our learning-based placement. Specifically, we first segment out the road region with Han et al. (2022) and sample placement locations in this region. To get a plausible orientation, we copy the

orientation of the closest car in the scene, assuming neighboring cars follow the same orientations. We used our proposed rendering pipeline to generate realistic augmentations.

**Lift-3D Li et al. (2023)** proposed a generative radiance field network to synthetize realistic 3D cars. Lift3D trains a conditional NeRF on multi-view car images generated by StyleGANs. However, the car shape is changed following the 3D bounding box dimensions. The generated cars are then placed on the road using a heuristic based on road segmentation. We used a single generated 3D car provided in the official code to augment the dataset as the training code is unavailable. Specifically, road region is segmented using off-the-shelf drivable area segmentor Han et al. (2022). Next, the 3D bounding box of cars is sampled from a predefined distribution of box parameters as given in Tab.8, and the ones outside the drivable area are filtered out. For a sampled 3D bounding box parameters b=$[b_x, b_y, b_z, b_w, b_h, b_l, b_\theta]$, we render the car at adjusted orientation angle $\tilde{\theta}$ using Eq. 7. We place the camera at the fixed height of $1.6m$, with an elevation angle of 0. Also, we used $(b_w, b_h, b_l)$ to render the car of a particular shape. We render the car image for 512x512 resolution using volume rendering and the defined camera parameters. Along with the RGB image, Lift3D also outputs the segmentation mask for the car which is used to blend it with the background. Fig. 12 shows some sample renderings from Lift3D.

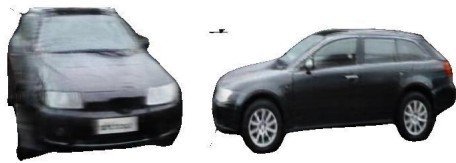

Figure 12: Sampled views rendered from Lift3D Li et al. (2023).

Table 8: Preset distribution of bounding boxes. Lift3D Li et al. (2023) samples bounding boxes from the predefined parameter distribution.

| Pose | Distribution | Parameters |
|---|---|---|
| $x$ | Uniform | $\{[-20m, 20m]\}$ |
| $y$ | Gaussian | $\mu = height, \sigma = 0.2$ |
| $z$ | Uniform | $\{[5m, 45m]\}$ |
| $l$ | Gaussian | $\mu = l_{\text{mean}}, \sigma = 0.5$ |
| $w$ | Gaussian | $\mu = w_{\text{mean}}, \sigma = 0.5$ |
| $h$ | Gaussian | $\mu = h_{\text{mean}}, \sigma = 0.5$ |
| $\theta$ | Gaussian | $\mu = \pm\pi/2, \sigma = \pi/2$ |

## A.4 ADDITIONAL OBJECT DETECTION RESULTS

### A.4.1 3D OBJECT DETECTION ON MONODETR ZHANG ET AL. (2022)

To validate the generalizability of our approach, we evaluate proposed 3D augmentation on a recent 3D monocular detection model MonoDETR Zhang et al. (2022) on the KITTI dataset in Tab. 9. We report the baseline results without our augmentation from the original paper. Our method consistently outperforms the baseline in

Table 9: Rendering ablation with fixed placement

| MonoDETR | 3D@IOU=0.7 | | | 3D@IOU=0.5 | | |
|---|---|---|---|---|---|---|
| | Easy | Mod. | Hard | Easy | Mod. | Hard |
| w/o 3D Augmentation | 28.84 | 20.61 | 16.38 | 68.86 | 48.92 | 43.57 |
| Geo-CP | 23.26 | 16.41 | 14.58 | 60.65 | 43.93 | 37.71 |
| Lift3D | 22.00 | 16.61 | 14.59 | 63.45 | 47.34 | 38.57 |
| RBP | 24.92 | 17.75 | 15.90 | 61.99 | 44.02 | 38.04 |
| Ours | **29.90** | **21.91** | **16.85** | **69.63** | **49.10** | **43.63** |

Table 11: Monocular 3D detection performance of Poisson Blending on our Rendering on KITTI Chen et al. (2015) validation set.

Table 12: MonoDLEMa et al. (2021) on Car with and without Poisson Blending

| Rendering | 3D@IOU=0.7 | | | 3D@IOU=0.5 | | |
|---|---|---|---|---|---|---|
| | Easy | Mod. | Hard | Easy | Mod | Hard |
| w/o 3D Aug. | 17.45 | 13.66 | 11.69 | 55.41 | 43.42 | 37.81 |
| Ours | **22.49** | **15.44** | **12.89** | **63.59** | **45.59** | **40.35** |
| Ours (+Poisson) | 21.34 | 14.44 | 12.81 | 59.60 | 44.11 | 38.15 |

Table 13: GUPNetLu et al. (2021) on Car with and without Poisson Blending

| Rendering | 3D@IOU=0.7 | | | 3D@IOU=0.5 | | |
|---|---|---|---|---|---|---|
| | Easy | Mod. | Hard | Easy | Mod | Hard |
| w/o 3D Aug. | 22.76 | 16.46 | 13.27 | 57.62 | 42.33 | 37.59 |
| Ours | **23.94** | **17.28** | **14.71** | **61.01** | **47.18** | **41.48** |
| Ours (+Poisson) | 22.43 | 17.03 | 14.55 | 60.00 | 45.28 | 39.60 |

all three settings. The comprehensive evaluation across several detectors (also in the main paper) evidently shows the generalization of our proposed 3D augmentation method.

### A.4.2 IMPROVING 2D OBJECT DETECTION

As our approach provides consistent 3D augmentations, it also enables to improve the performance of 2D object detectors. Specifically, our placement model also predicts the 2D bounding box along with the 3D bounding box (followed in most of the 3D detection works). We use these predicted 2D bounding box annotations to obtain a labeled 2D detection dataset. We evaluate the gains from our augmentations on 2D object detection on off-the-shelf

Table 10: 2D Detection Performance on '*Car*' category with CenterNet Zhou et al. (2019)

| Rendering | AP2D@IOU=0.5 | | |
|---|---|---|---|
| | Easy | Mod. | Hard |
| w/o 3D Aug. | 86.03 | 73.74 | 65.08 |
| Ours | **89.56** | **76.79** | **72.28** |

2D detector CenterNet Zhou et al. (2019) in Tab. 10. Following Simonelli et al. (2019b), we use a standardized approach to report $AP_{40}$ metric instead of the $AP_{11}$ for evaluation. Notably, our proposed augmentation method, though designed for 3D detection, can also improve the performance of 2D object detection, proving the task generalization of the proposed approach.

### A.4.3 EFFECT OF POISSON BLENDING

We use Poisson blending to enhance the quality of the composition of synthetic cars with the background scene. We observe a slight dip in the detection performance using the obtained augmentations as reported in Tab. 11. A similar observation was made in Zhao et al. (2023), where improved blending does not positively affect the detection performance.

### A.5 DATA EFFICIENCY

In this section we demonstrate the data efficiency of our method. As observed in Tab.14 our method can significantly reduce the dependence on real data when training

Table 14: Data efficiency of SA-PlaceNet

| MonoDLE | | 3D@IOU=0.7 | | | 3D@IOU=0.5 | | |
|---|---|---|---|---|---|---|---|
| % Real Data | % Aug. Data | Easy | Mod. | Hard | Easy | Mod. | Hard |
| 10 | 10 | 4.94 | 3.90 | 3.26 | 27.21 | 21.03 | 18.06 |
| 25 | 25 | 13.38 | 9.78 | 8.23 | 48.28 | 36.99 | 30.83 |
| 50 | 50 | 20.46 | 13.70 | 11.71 | 58.04 | 43.83 | 37.87 |
| 75 | 75 | 21.53 | 14.95 | 12.38 | 60.94 | 45.19 | 39.99 |
| 100 | 100 | **22.49** | **15.44** | **12.89** | **63.59** | **45.59** | **40.35** |
| 100 | 0 | 17.45 | 13.66 | 11.69 | 55.41 | 43.42 | 37.81 |

monocular detection networks. Specifically augmenting just 50 % of the real data can achieve better performance than training with 100 % of original training data.

## A.6 RENDERING CARS

### A.6.1 COPY-PASTE

We provide details about a simple copy-paste rendering, where the cars from the training corpus are added to the predicted 3D bounding boxes. We extract

cars of various orientations from the training set images through instance segmentation using Detectron2 Wu et al. (2019). These cars are archived in a database with their corresponding 3D orientation and binary segmentation mask data. During inference, given a 3D bounding box, we query and search for cars whose orientation closely aligns with the given 3D box orientation. A certain degree of randomness is introduced in selecting the nearest-matching car, contributing to increased diversity and seamless integration with the input scene. Next, we compose the retrieved car image onto the background scene using the 2D-coordinated obtained from the 3D bounding box and the binary mask. This simple rendering essentially captures the diverse cars present in the training dataset and helps in generating scenes that are close to training distribution. However, such rendering has a problem with shadows as the composition is not 3D-aware given the placed cars are stored as images.

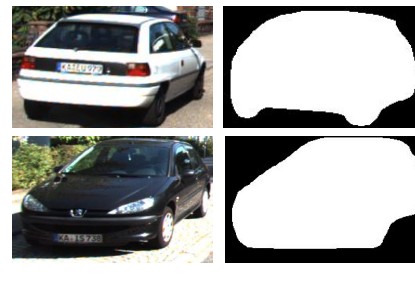

Copy–Paste Car      Binary Mask

Figure 13: Sample cars from the Copy-Paste Database

### A.6.2 SHAPENET

ShapeNet Chang et al. (2015) is a large-scale synthetic dataset that provides 3D models for various object categories, including cars. The ShapeNet Cars dataset focuses specifically on providing 3D models of different car models from various viewpoints. We leverage the high diversity of cars (nearly 7500 models) in the dataset and render the cars at

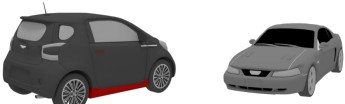

Figure 14: Sample of ShapeNet Chang et al. (2015) cars rendered at different views.

the predicted box locations with 3D bounding box parameters using Blender Community (2018) software. We employ a random sampling technique to select a 3D car model from this extensive dataset, which is then loaded in the Blender Community (2018) environment. To ensure consistency in the car shapes, we initially calculated the average dimensions of the cars within the dataset. We exclude any car model with dimensions exceeding $50\%$ of the computed average, and we repeat this random sampling procedure until the specified conditions are satisfied. Following that, we align and render the car by a 3D rotation angle. Specifically, as the orientation angle $\theta$ is defined in 3D, using it directly to render the image does not take care of perspective projection. Eg. all the cars following a lane will have similar orientation angles (close to zero) but look visually different when projected on the image as shown in Fig. 15. Both the rendered cars have $0$ orientation angle in 3D but when projected onto the image planes, the rendered orientation changes with the location. To this end, we adjust the car orientation by a correction factor to incorporate the perspective view, as described in equation equation 7,

$$\tilde{\theta} = \theta + \tan^{-1}\left(\frac{x}{z}\right) \qquad (7)$$

where $x$ and $z$ are the respective 3D coordinates of the bounding box. We use the final corrected $\tilde{\theta}$ value for rendering the ShapeNet car. We render car images at $512\text{x}512$, with a white background, which can be later used as a segmentation mask to blend the rendered image. A few examples of the ShapeNet cars rendered with different orientations are visualized in Fig. 14.

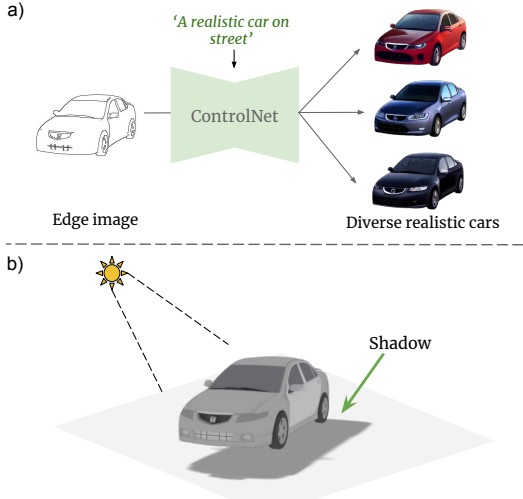

Figure 15: Perspective and Absolute projection of cars with the same 3D orientation.

## A.7 REALISTIC RENDERING USING TEXT-TO-IMAGE MODEL

### A.7.1 CONTROLNET ZHANG & AGRAWALA (2023) BASED RENDERING.

We leverage a state-of-the-art image-to-image translation method to convert the synthetic ShapeNet renderings into realistic cars that blend well with the background scene. We use edge-conditioned ControlNet, which takes an edge image and a text prompt to generate images following the edge map and the prompt. Specifically, we utilize an edge detector to create edge maps for synthetic car images rendered using ShapeNet Chang et al. (2015), preserving the car's structure while maintaining its original orientation and scale. These edge maps, generated through the Canny Edge Detection algorithm Canny (1986), serve as input for the edge-conditioned ControlNet Zhang & Agrawala (2023), enabling the rendering of realistic cars using the prompt *'A realistic car on the street'*. Furthermore, given an edge map and hence a ShapeNet-rendered car, we can obtain various realistic renderings at each iteration, facilitating diverse scene generations (Fig. 16). We further enhance ControlNet's backbone diffusion model using LoRA Hu et al. (2022) on a subset of 'car' images from the KITTI dataset. This process enables the generation of natural-looking car versions that seamlessly blend with the background scene. Finally, we integrate the ControlNet-rendered car and its shadow base into the predicted location within the scene to achieve a realistic rendering.

Figure 16: Diverse renderings generated with edge-conditioned ControlNet.

### A.7.2 RENDERING REALISTIC SHADOWS.

Shadows are realistically generated using the ShapeNet Chang et al. (2015) Cars dataset and rendered with Blender Community (2018) software, following the rendering procedure outlined in A.6.2. However, to generate shadows, we modify the rendering method by introducing a 2D mesh plane beneath the car base and adding a uniform 'Sun' Light source along the z-axis of the blender environment, placed in the top on the z-axis of the car (Fig. 16). Additionally, we introduce slight variations across all axes for the light source position. Once the cars are positioned within the

Blender Community (2018) environment with suitable orientation, we render the entire scene while setting both the car and the 2D plane as transparent. This method enables us to create a collection of shadow renderings with a transparent background for each car in the placement setting.

