# OpenReview forum: "Smart Placement Enhanced Vision: Enhancing 3D-Detection With Learned 3D Placement"
_ICLR.cc/2025/Conference — ICLR 2025 Conference Withdrawn Submission_

### Official Review · Reviewer_tmmy · 2024-10-31

**Soundness:** 3
**Presentation:** 3
**Contribution:** 2
**Rating:** 3
**Confidence:** 4

**Summary:**

Summary:

They propose a a novel scene-aware augmentation method (serve as an effective 3D augmentations) for 3d object detection task.

1. 3D Scene-Aware Placement Network (SA-PlaceNet) is proposed to maps a given scene image to a distribution of plausible 3D bounding boxes. It learns realistic object placements that adhere to the physical rules of road scenes.

2. A rendering pipeline with ControlNet is presented to leverage synthetic 3D assets and an image-to-image translation model.

3. A two-stage augmentation approach is designed, which allows seamless integration with advancements in placement and renders for enhancing 3D object detection datasets.

4. Extensive ablation studies show the effectiveness of the proposed method.

**Strengths:**

Strengths:

1. Describtions of motivation and problem analysis as well as insight dicussion are good.
2. Method pipeline is clear and easy to understand.
3. Extensive visulization of intermediate results are provided.

**Weaknesses:**

Weaknesses:

1. Most of technique details seem to be incremental and full with engineering details. ICLR is a conference which mostly focuses on learning theory and related topics.
2. Rendering results are not realistic. For example, no shadow.
3. Eq. (5) is the core of the proposed method. However, the detail of loss function is not defined clearly. And we do not know why this loss function can solve the problem of bounding box distribution.
4. Experiments on other datasets are insufficient.

**Questions:**

Most of technique details seem to be incremental and full with engineering details. ICLR is a conference which mostly focuses on learning theory and related topics.

Eq. (5) is the core of the proposed method. However, the detail of loss function is not defined clearly. And we do not know why this loss function can solve the problem of bounding box distribution.

---

### Official Review · Reviewer_UfXU · 2024-11-03

**Soundness:** 2
**Presentation:** 3
**Contribution:** 2
**Rating:** 3
**Confidence:** 4

**Summary:**

This paper introduces Smart-Placement, a novel approach to data augmentation for 3D object detection from RGB images. The key insight is that for realistic augmentations, not only does the rendered object need to look good (object appearance), but it needs to be placed naturally in the scene (object placement). To do this, the authors train a network to learn where objects could plausibly exist in a scene using supervision data with ground-truth 3D boxes, then they use this to guide object placement when augmenting training data with synthetic cars. The method proves to be effective, improving performance of existing 3D detectors on the KITTI and NuScenes datasets, especially when considering the training efficiency.

**Strengths:**

- The paper is well structured and clearly describes two problems it's trying to tackle: object placement and object appearance for data augmentation. I agree with the premise that synthetic object insertion could help the performance of downstream tasks, especially 3D prediction tasks where it's hard and expensive to obtain annotations for supervised learning. The technical details are clearly described and easy to follow.

- The method shows good performance on monocular 3D object detection benchmarks, especially when it comes to training data efficiency.

**Weaknesses:**

The paper has several major issues:

* Big issue with problem setting: although I agree that data augmentation will help downstream tasks, usually the motivation for data augmentation is to cover "edge cases" - cases where we lack training data. With the method described in this paper, this is not the case: the network will simply learn to mimic the distribution of the training data (with some small wiggle room). This is shown in Figure 5, where the distribution of training orientations is almost the same as the distribution of generated orientations. Therefore, it's unclear what the actual application the paper is looking at, and where the quantitative improvements come from (I speculate that maybe the testing orientations are also very similar to the training orientations, leading to some form of overfitting, but more analysis is needed here).

* Lack of novelty in general, with some questionable assumptions: the procedure for "geometric-aware augmentation" doesn't seem that much different from Rule-based Placement, which is also based on heuristics for how to place cars. These heuristics are based on assumptions that I don't think always hold true, e.g., L287 "the regions' neighboring ground truth car locations are also plausible for placement". This could be true in highway settings with simple straight-line roads, but this procedure cannot guarantee to generate cars with plausible placements on the road.

* Some claims are not backed up or are not correct. For example, the paper claims that "the cars in our rendering match the scene lighting conditions well". Based on the rendering pipeline in Figure 3, since the background scene is never considered/conditioned in the ControlNet process, at best we will get a realistic car but with arbitrary lighting. So, it's unclear why the rendered cars could match the scene lighting conditions well. This is one of the claimed contributions in the paper, but these claims are shaky and more explanation/analysis is needed.

**Questions:**

See Weaknesses. I encourage the authors to revisit and revise some of the strong claims in the paper and if possible, back them up with reasonable evidence to convince the reviewers. I'm also open to discussion about the 1st point in Weaknesses (issue with problem setting) as I could be missing something here.

---

### Official Review · Reviewer_ZTQx · 2024-11-04

**Soundness:** 2
**Presentation:** 2
**Contribution:** 2
**Rating:** 5
**Confidence:** 4

**Summary:**

This paper proposes Smart-Placement, a 3D scene aware augmentation method for generating diverse and realistic augmentations, to improve 3D detection performance.

**Strengths:**

- The idea of learning the location distribution is interesting and reasonable.

- The experiments on KITTI val set  is extensive, showing the effectiveness of the overall pipeline.

**Weaknesses:**

-  3D-aware object placement and realistic appearance is important in data augmentation and  data augmentation is important for 3D detection. It is a common sense, thus I think Contribution-1 is over-claimed.

- KITTI is a small dataset, the test set results are important

- More fine-grained ablations are missed. e.g., the ablation of location distribution and the choice of rendering.

- Some other highly-related based methods should be discussed, e.g. [1].


[1] Exploring Data Augmentation for Multi-Modality 3D Object Detection.

**Questions:**

Please see the weakness.

---

### Official Review · Reviewer_aA7r · 2024-11-04

**Soundness:** 2
**Presentation:** 2
**Contribution:** 2
**Rating:** 3
**Confidence:** 5

**Summary:**

The paper proposes **Smart-Placement**, a novel approach for scene-aware augmentation in monocular 3D object detection, particularly enhancing the placement realism of augmented objects within 3D scenes. Utilizing a Scene-Aware Placement Network (SA-PlaceNet), it learns to generate 3D bounding box distributions that respect realistic scene constraints, such as object location, and orientation. This network allows plausible and physically consistent placement of synthetic objects, which are then rendered using a diffusion model, creating realistic scene augmentations. Extensive evaluations on the KITTI and NuScenes datasets show that Smart-Placement significantly improves monocular 3D object detection performance, surpassing previous state-of-the-art methods and achieving comparable performance with less data. This approach is modular and data-efficient, demonstrating high adaptability for autonomous driving datasets.

**Strengths:**

- The paper introduces a novel distribution-based placement network, going beyond simple heuristic placements, resulting in highly plausible object positioning.

- The method is evaluated on both KITTI and NuScenes datasets, showcasing clear performance gains for monocular 3D detection.

**Weaknesses:**

1. While promising, the method may not fully align with recent advances in autonomous driving perception. Bird’s-eye view detection has shown significant potential in this area, and multi-view reconstruction approaches, based on NeRF and 3DGS[1, 2], can provide even more realistic object placements, potentially making this approach appear comparatively less advanced.
2. The proposed pipeline relies on multiple pre-trained networks for tasks like inpainting, depth prediction, and style transfer, which may introduce additional sources of uncertainty and inaccuracy in the final scene augmentations.
3. Lighting inconsistencies exist, as the predefined lighting in the augmented images does not always align well with the original scene’s lighting conditions, potentially detracting from the realism (as shown in Figure 4).
4. Object insertion could be further refined, as car image are conditioned solely on 3D asset edge images without incorporating cues from the surrounding scene, resulting in an artificial appearance in some cases (e.g., Figure 9).
5. The typesetting and formatting could be improved; the excessive use of `\vspace` creates a disjointed layout that could benefit from further refinement to enhance readability.

[1] Tonderski et al. NeuRAD: Neural Rendering for Autonomous Driving. CVPR 2024

[2] Yan et al. Street Gaussians: Modeling Dynamic Urban Scenes with Gaussian Splatting. ECCV 2024

**Questions:**

See Weaknesses.

---

### Note · Authors · 2024-11-14

**Comment:**

I wish to withdraw this submission from ICLR.

**Withdrawal Confirmation:**

I have read and agree with the venue's withdrawal policy on behalf of myself and my co-authors.